# Nanoformulation of Seaweed *Eisenia bicyclis* in Albumin Nanoparticles Targeting Cardiovascular Diseases: In Vitro and In Vivo Evaluation

**DOI:** 10.3390/md20100608

**Published:** 2022-09-27

**Authors:** Sofia Pinto, Maria Manuela Gaspar, Lia Ascensão, Pedro Faísca, Catarina Pinto Reis, Rita Pacheco

**Affiliations:** 1Departamento de Engenharia Química, Instituto Superior de Engenharia de Lisboa (ISEL), Avenida Conselheiro Emídio Navarro 1, 1959-007 Lisboa, Portugal; 2Research Institute for Medicines (iMed. ULisboa), Faculty of Pharmacy, Universidade de Lisboa, Avenida Professor Gama Pinto, 1649-003 Lisboa, Portugal; 3Centro de Estudos do Ambiente e do Mar (CESAM), Faculdade de Ciências, Universidade de Lisboa, Campo Grande, 1749-016 Lisboa, Portugal; 4Faculdade de Medicina Veterinária, Universidade Lusófona de Humanidades e Tecnologia, 1749-024 Lisboa, Portugal; 5CBIOS-Research Center for Biosciences and Health Technologies, Universidade Lusófona de Humanidades e Tecnologia, 1749-024 Lisboa, Portugal; 6Instituto de Biofísica e Engenharia Biomédica, Faculdade de Ciências, Universidade de Lisboa, Campo Grande, 1749-016 Lisboa, Portugal; 7Centro de Química Estrutural, Institute of Molecular Sciences, Universidade de Lisboa, 1749-016 Lisboa, Portugal

**Keywords:** *Eisenia bicyclis*, cholesterol, cardiovascular diseases, nanoparticles, in vitro and in vivo evaluation

## Abstract

Natural products, especially those derived from seaweeds, are starting to be seen as effective against various diseases, such as cardiovascular diseases (CVDs). This study aimed to design a novel oral formulation of bovine albumin serum nanoparticles (BSA NPs) loaded with an extract of *Eisenia bicyclis* and to validate its beneficial health effects, particularly targeting hypercholesterolemia and CVD prevention. Small and well-defined BSA NPs loaded with *Eisenia bicyclis* extract were successfully prepared exhibiting high encapsulation efficiency. Antioxidant activity and cholesterol biosynthesis enzyme 3-hydroxy-3 methylutaryl coenzyme A reductase (HMGR) inhibition, as well as reduction of cholesterol permeation in intestinal lining model cells, were assessed for the extract both in free and nanoformulated forms. The nanoformulation was more efficient than the free extract, particularly in terms of HMGR inhibition and cholesterol permeation reduction. In vitro cytotoxicity and in vivo assays in Wistar rats were performed to evaluate its safety and overall effects on metabolism. The results demonstrated that the *Eisenia bicyclis* extract and BSA NPs were not cytotoxic against human intestinal Caco-2 and liver HepG2 cells and were also safe after oral administration in the rat model. In addition, an innovative approach was adopted to compare the metabolomic profile of the serum from the animals involved in the in vivo assay, which showed the extract and nanoformulation’s impact on CVD-associated key metabolites. Altogether, these preliminary results revealed that the seaweed extract and the nanoformulation may constitute an alternative natural dosage form which is safe and simple to produce, capable of reducing cholesterol levels, and consequently helpful in preventing hypercholesterolemia, the main risk factor of CVDs.

## 1. Introduction

According to the World Health Organization (WHO), cardiovascular diseases (CVDs) are the leading cause of mortality in both industrialized and developing countries. By 2019, about 17.6 million people died from CVDs, representing 32% of all global deaths, with 85% of these deaths being due to heart attack and stroke [1].

The scientific community and the population are increasingly aware of the importance of CVD prevention. These pathologies are associated with healthy lifestyle habits such as a nutritious diet [2] and are addressed by treating major risk factors, such as hyperlipidemia and hypercholesterolemia [3]. Nevertheless, current drug therapies for hypercholesterolemia have considerable adverse side effects, such as muscle pain or hepatotoxicity [4,5,6]. Consequently, new effective and safer strategies are demanded. Several natural products, mainly of plant and algae origin, have been referenced for their potential in the prevention and treatment of various diseases [7], namely CVDs [8].

The demand for algae, whether for human consumption or for industrial processing, has increased in the last few decades. Though these are considered abundant natural resources, seaweed cultivation is providing sustainable biomass availability; by 2010, the algae worldwide production reached 15.8 million tonnes [9].

*Eisenia bicyclis* (Kjellman) Setchell (Laminariaceae), commonly named as Arame, is a perennial brown seaweed distributed along the mid-Pacific coastlines of Korea and Japan that is traditionally consumed in the diet or used in food-related products [10,11,12,13,14]. The main bioactive compounds identified in this seaweed are phenolic compounds, such as phlorotannins (e.g., eckol) [15], polysaccharides (e.g., laminarin) [16], and carotenoids (e.g., fucoxanthin) [17]. As with other edible marine seaweeds, *E. bicyclis* is mainly consumed in a traditional mode in the form of food or extracts. These dosage forms are very complex mixtures of bioactive compounds showing different physical/chemical properties, such as solubility, and stability limitations. Consequently, nanotechnology may benefit its potential of application, namely as food supplements or drugs.

The application of nanotechnology to drugs or nutraceuticals is an emerging area, particularly in the design and production of appropriate nanosized bioactive delivery systems offering several advantages, such as optimal bioactive release and loading properties, and further optimizes the therapeutic efficacy of the encapsulated material [18].

Protein nanoparticles (NPs) are successful examples of nanoformulations that can stabilize synthetic drugs [19,20] as well as extracts from plants [21]. Protein-based NPs are highly biodegradable and non-toxic, showing non-antigenicity, with the advantage of being easy to prepare and monitor in size while also capable of covalently binding to encapsulated material and/or ligands, thus offering the possibility of surface modification [22,23]. Among the variety of proteins commonly used to prepare NPs, bovine albumin serum (BSA) has been widely used because, apart from its biocompatibility and biodegradability [23,24], BSA has the ability to self-associate while also presenting emulsifying, foaming, and gelation properties [25]. Overall, it is an appropriate candidate for the encapsulation of both hydrophilic and hydrophobic bioactive ingredients, such as phenolic compounds, carotenoids, polyunsaturated fatty acids, and vitamins. It is also the most abundant protein in blood plasma, exhibiting active targets and specific activity in the liver–pancreatic system [26] and exhibiting high half-life time in the bloodstream (19 days) [27]. All these advantages make BSA suitable for NP preparation.

The production of BSA NPs containing an aqueous extract of *E. bicyclis* (EEB) is herein reported and their biological activities and potential health benefits, particularly in the reduction of plasma cholesterol, were evaluated. When a nanoproduct is developed, a broad range of parameters must be attained regarding the required characteristics of safety, efficacy, improved bioactive delivery, and applicability [18]. Therefore, safety and permeation across the biological barriers of the NPs in an ex vivo model of the intestinal lining were assessed. In addition, the effect of the nanoformulation’s administration on the metabolism of Wistar rats was also reported. Although some data regarding the encapsulation of marine algae extracts have been described in the literature, to the best of our knowledge no studies using seaweed *E. bicyclis* have been published to date. Therefore, the present work using BSA NPs loaded with an aqueous extract of this seaweed represents a novel approach against hypercholesterolemia. Furthermore, the biological effects, as well as the underlying mechanisms of action, were also investigated.

## 2. Results

### 2.1. Preparation and Characterization of the E. bicyclis Extract (EEB)

A re-emerging interest in the use of natural products for industrial purposes has been observed due to the reputable sources of new active pharmaceutical ingredients that the natural products provide [21]. *E. bicyclis* is often consumed in the diet or used in food-related products [11] and has several identified bioactive compounds [15,16,17]. However, the effect of *E. bicyclis* extract (EEB) in relation to hypercholesterolemia is still poorly studied. Therefore, the aim of this study was to provide a scientific support and validation for the use of EEB against hypercholesterolemia.

From a commercial *E. bicyclis* algae biomass, an aqueous extraction was carried out with a 66% yield. The aqueous extraction, besides being environmentally friendly, may simulate seaweed product intake habits, usually in the form of soups or beverages, thus allowing for better assessment of the beneficial effects associated with the consumption of EEB.

Following this, the quantification of total phenols, polysaccharides, and total protein in EEB was performed. A total phenolic content of 0.062 ± 0.005 mg of phloroglucinol equivalents/mg was observed in EEB. Total polysaccharides of 0.221 ± 0.002 mg of polygalacturonic acid equivalents/mg was achieved in EEB. The protein level in EEB was seen to be very low, below the method’s detection limit (<0.5 µg).

### 2.2. Preparation and Characterization of the EEB-Loaded BSA NPs

Nanotechnology, through the development of NPs, has been revolutionizing drug delivery, since its encapsulation significantly improves the bioactive release, absorption, and safety profile of loaded compounds or extracts [28,29]. Despite its success, the formulation of NPs containing bioactive compounds from seaweeds is still a largely unexplored topic [30]. Herein, EEB-loaded BSA NPs were prepared and studied both in vitro and in vivo.

The BSA NPs were prepared as reported by Santos-Rebelo et al. [31]. Table 1 displays the main results for the characterization of the prepared nanoformulations. The developed BSA NPs presented a mean size ranging from 71 to 226 nm, with polydispersity index (PdI) values from 0.285 to 0.713. All BSA NPs prepared exhibited a slightly negative surface charge, ranging from −11 to −16 mV. The encapsulation efficiency (EE) of the EEB in BSA NPs was evaluated using different initial EEB amounts, namely 10, 25, and 50 mg. It was found that BSA NPs with 10 mg of EEB showed the highest EE, with a value of 96%. Considering these results, mainly the mean size distribution, the initial amount of free extract selected to be encapsulated into BSA NPs was 10 mg. The production yield of EEB-loaded BSA NPs (10 mg) was 70% and this was evaluated in further testing.

The morphology of the BSA NPs, assessed by scanning electron microscopy (SEM), showed that the empty BSA NPs (Figure 1a) and the EEB-loaded BSA NPs (Figure 1b) presented a generally spherical shape with a smooth surface. However, the empty BSA NPs showed a more uniform distribution with scarce agglomerates, confirming dynamic light scattering (DLS) results.

### 2.3. Biological Activities of E. bicyclis Extract (EEB) in the Free Form and after Encapsulation in BSA NPs (EEB NPs)

#### 2.3.1. DPPH Radical Scavenging Assay

Antioxidant activity (AA) was determined using the 2,2-diphenyl-1-picrilhidrazil (DPPH) scavenging method. Figure 2 shows the main results. It was observed that the antioxidant activity of EEB decreased after encapsulation into BSA NPs. Free EEB showed an antioxidant activity of 65 ± 3%, while for EEB-loaded BSA NPs (EEB NPs) 8 ± 1% was achieved. This decrease in antioxidant activity, after extract encapsulation, may be due to the fact that the bioactive compounds responsible for the AA from the extract are poorly exposed at the surface of NPs [21], or may even due to strong interactions between the bioactive compounds and BSA [32], limiting the scavenging effect as the extract compounds in the NPs become less available for the reduction of the DPPH radical during assay time.

#### 2.3.2. Inhibition of Acetylcholinesterase (AChE) Enzyme Activity

CVDs are associated with an increased risk of developing neurodegenerative diseases [33,34]. Several dietary components, some of which are from marine sources, are known to delay the progression of cardiovascular and neurodegenerative diseases such as cognitive decline and Alzheimer’s Disease (AD).

Acetylcholinesterase (AChE) inhibitors, such as donepezil (a neuroprotective drug used to ameliorate cognitive issues in AD) [35] and others that stimulate gastrointestinal motility, are often associated with several side effects [36]. The capacity of the natural compounds from EEB, in the free form or loaded in BSA NPs (EEB NPs), to inhibit AChE was evaluated and the results are shown in Figure 3.

From the results shown in Figure 3, it can be concluded that, at a concentration of 0.25 mg/mL, EEB has the ability to inhibit the AChE enzyme; however, at the same concentration, the nanoformulation (EEB NPs) showed enhanced enzyme inhibition, with the encapsulated extract having a higher capacity to inhibit this enzyme than its free form. It is suggested that this improvement from extract encapsulation may be used with a neuroprotective effect and to increase gut motility, as has already been seen in rats [36].

#### 2.3.3. Effects against Hypercholesterolemia—HMGR Inhibition and Reduction of Cholesterol Permeation

Hypercholesterolemia is characterized by high levels of total cholesterol (Tc), low-density lipoprotein cholesterol (LDL-c), or triglycerides (TGs) and/or a decrease in high-density lipoprotein cholesterol (HDL-c) [37]. An imbalance between de novo biosynthesis and dietary cholesterol absorption is the main cause that leads to hypercholesterolemia and the drug targets for this pathology [38]. Other natural compounds from marine sources have demonstrated an anti-hypercholesterolemic effect during the search for alternative therapies or functional foods [39]; however, little is known about *E. bicyclis* and its mechanism of action [40].

In order to evaluate if the prepared EEB, in free or nanoformulated forms (EEB NPs), were able to reduce blood cholesterol levels and consequently prevent CVDs, two approaches were addressed similarly to the current therapies: the evaluation of (1) 3-hydroxy-3-methylglutaryl-CoA reductase (HMGR) inhibition, the main liver endogenous de novo cholesterol biosynthesis enzyme, and (2) the reduction in cholesterol intestinal permeation using a model of human Caco-2 cells simulating the intestinal lining.

Figure 4 shows the results of enzyme HMGR inhibition by encapsulated EEB in BSA NPs (EEB NPs) in comparison to the extract free form (EEB) and pravastatin, a statin drug (positive control) often prescribed for hypercholesterolemia treatment. From the results it was possible to observe that the EEB, both free and encapsulated in the nanoformulation (EEB NPs), showed a high capacity to inhibit this enzyme within the range of pravastatin, a commercial and clinically used drug. A brief literature search showed that this is the first report demonstrating the capacity of EEB and its nanoformulation to inhibit HMGR. These results support the potential use of both the extract and nanoformulation against hypercholesterolemia via the same mechanism of inhibition as the prescribed drug, also validating the claimed health effects of *E. bicyclis* seaweed (decreasing CVD risk).

The effect of both free extract (EEB) and EEB-loaded BSA NPs (EEB NPs) for reducing cholesterol permeation in the intestinal lining model of the Caco-2 cell line, which differentiates into a functional and morphologically similar state to human enterocytes, was additionally evaluated [41]. After 6 h of cell incubation with 5 mM of cholesterol, cholesterol permeation to the basolateral chamber in the transwell insert was determined and considered as 100% permeation. The capacity of the extract (EEB) and nanoformulation (EEB NPs) to affect cholesterol permeation in the cell model is represented in Figure 5 as the percentage of the reduction of cholesterol permeation. The results obtained were compared to ezetimibe (Ezet) alone, or in combination, as positive control. Ezetimibe is a drug that is prescribed for hypercholesterolemia which blocks dietary exogenous cholesterol absorption through the gastrointestinal barrier [42].

From the results shown in Figure 5, it can be seen that the extract, EEB, reduces cholesterol permeation by 32% and that the simultaneous use of the extract and ezetimibe (EEB + Ezet) causes the effect on the reduction of cholesterol permeation to be potentiated, achieving a 42% reduction (which is a value within the range of the effect of ezetimibe (Ezet)). This same effect was obtained using the nanoformulations, EEB-loaded BSA NPs, either in the absence (EEB NPs) or presence of ezetimibe (EEB NPs + Ezet). An increased reduction in cholesterol permeation was particularly seen for the latter, relative to the effect of sole use of the drug (Ezet).

Given the previous results, it seems that encapsulation of the extract in BSA NPs intensifies its efficacy for reducing cholesterol permeation in the intestinal lining and is even more effective if used in combination with ezetimibe therapy. These results further suggest the potential of the extract, especially as extract-loaded BSA NPs, for hypercholesterolemia treatment.

### 2.4. Bioactive Compound Bioavailability in Intestinal Lining Model Caco-2 Cells

The Compounds identified in EEB are mostly phlorotannins, consisting of several units of phloroglucinol [13,43]. The amount of phloroglucinol derivatives present in the extract was quantified by the HPLC-DAD as being approximately 0.3% (m/m) (data not shown). Considering that intestinal permeability is a limiting factor for the oral absorption and bioavailability of some drugs, the permeation of the extract compounds in the intestinal lining model Caco-2 cells was evaluated in order to verify if EEB or EEB-loaded BSA NPs (EEB NPs) were consumed in the presence of cholesterol and ezetimibe, i.e., if its compounds could permeate the gastrointestinal barrier.

From the results obtained (Figure 6), it can be concluded that between 20–30% of the compounds present in the EEB and EEB-loaded BSA NPs crossed the gastrointestinal barrier.

### 2.5. In Vitro Safety Assay

Cytotoxicity against human intestinal Caco-2 cells and liver HepG2 cells was assessed in order to establish the safety profile of EEB in free and nanoformulated forms.

Figure 7 shows the results obtained after 24-h incubation of both cell lines with EEB and EEB-loaded BSA NPs. It was seen that, at an extract concentration of 1 mg/mL, a cell viability above 80% was achieved in all cases. According to Biok’s criteria, the four categories of extract cytotoxicity are (regarding the value of extract concentration decreasing 50% of cell viability (IC_50_)) very active (IC_50_  ≤  20 μg/mL), moderately active (IC_50_  >  20–100 μg/mL), weakly active (IC_50_  >  100–1000 μg/mL), and inactive (IC_50_  >  1000 μg/mL) [44]. Thus, both the extract and nanoformulation can be considered inactive, with no associated cytotoxic activity implying they are therefore safe according to the above criteria.

### 2.6. In Vivo Efficacy and Safety Assay

#### 2.6.1. First Set of the In Vivo Assay: *Eisenia bicyclis* Free Extract (EEB)

Following the in vitro safety results on human cell lines, in vivo safety, as well as the efficacy of free EEB, was firstly evaluated in Wistar rats after oral administration via drinking water over a period of 5 weeks (first set of experiments) of either the extract, ezetimibe, or both. Each rat, with an average body weight of 390 g, consumed about 50 mL of water per day; in the supplemented groups, the daily concentration of the extract or ezetimibe in the drinking water was 0.1 mg/mL. Animal body weight (b.w.) (Table 2), feed intake (Table 3), and blood glucose (Table 4) were checked weekly. The mortality rate was also evaluated.

The results suggest a non-significant effect from the supplementation of the extract (EEB), ezetimibe (Ezet), or both substances (EEB + Ezet) on the animal’s body weight (b.w.) over the 5 weeks of the assay (Table 2); therefore, the seaweed extract, seen to have very low protein content and typically containing trace lipids and fatty acid content [45,46], brought no major increase in calories to the fed water.

Regarding feed intake (Table 3), it was seen, by week 3, that the extract-supplemented groups (EEB and EEB + Ezet) consumed less feed compared to the control and ezetimibe (Ezet) groups, which suggested that the extract-supplemented groups may experience enhanced satiety when compared to the other groups.

The results also demonstrated that there was no significant difference in the rat’s blood glucose (Table 4) over the duration of this assay. In the group supplemented with ezetimibe (Ezet), a small decrease in blood glucose was seen by week 3 and 4. Although this decrease could be in agreement with the ability of ezetimibe to reduce blood glucose that is often described in the literature [47], this was seen to not be significant by week 5.

Other serological parameters, such as cholesterol and urea concentrations, were assessed for the different groups in order to determine the influence of the administration of the extract (EEB), ezetimibe (Ezet), and both substances (EEB + Ezet) on the cholesterol levels of Wistar rats and their renal function. The determination of urea is of considerable importance, as it is a parameter used for monitoring the metabolic activity of the body, especially regarding the normal functioning of the kidney.

In the beginning of this first set of animal experiments, it was observed that the plasma cholesterol concentration of the rats was 40 mg/dL, in agreement with the reference values described in the literature [48]. From Figure 8, it can be seen that significant differences between groups, in terms of cholesterol values, were only observed for animals supplemented with extract and ezetimibe (EEB + Ezet) when compared to the control group, with only this group showing a decrease in cholesterol levels. It is also possible to conclude, as seen in Figure 8, that the urea values did not show to be significantly different between the animal groups and that the values are in line with those expected for Wistar rats with normal metabolism [49]. Therefore, it was seen that administration of the extract, ezetimibe, and the mixture of both did not compromise the renal function of the rats in the first set of the in vivo assay.

#### 2.6.2. Second Set of the In Vivo Assay: Free EEB versus EEB-Loaded BSA NPs

To evaluate the safety and efficacy of the prepared nanoformulation, a second set for the in vivo assay was developed. In this set, 20 male Wistar rats were separated in different groups and supplemented 5 days a week for 4 weeks through oral gavage with 1 mg per day of extract, either in free form (EEB) or loaded to BSA NPs (EEB NPs), or with both extract and ezetimibe (EEB + Ezet), which showed the best reduction of blood cholesterol in the first set of experiments. These groups were compared to the non-supplemented rat control group.

Results on body weight are depicted in Table 5. Again, no differences, as in the first set of experiments in body weight (b.w.), were observed in this trial.

Considering the data in Table 6, results showed that there were only slight changes in the feed intake of the rat groups during the trial. However, these oscillations were more noticeable for the rats supplemented with both the extract and ezetimibe (EEB + Ezet) and EEB-loaded BSA NPs (EEB NPs), where an improvement in feed intake was observed that is contrary to the previous findings for the first trial involving extract and ezetimibe supplementation of drinking water. However, no weight gain could be associated to this small increase.

The results regarding blood glucose shown in Table 7 show that, in week 4, most of the animals showed higher values of glycemia; however, all these results are within the reference values and therefore were not considered significant. Furthermore, the highest observed blood glucose values may be due to the fact that, at the end of the experimental protocol, the stress in the animals may lead to higher blood glucose values [50].

Cholesterol and urea concentrations, for evaluation of renal function, were determined for the different groups under study in this trial (Figure 9). The results showed that, in all supplemented groups, the cholesterol level was reduced relative to the control. It was observed that both the free form extract (EEB) and EEB-loaded BSA NPs (EEB NPs) showed efficacy in reducing cholesterol in vivo and were within the range of the combination of both the extract and ezetimibe (EEB + Ezet) during the 4-week duration of the assay.

From Figure 9, it was also possible to conclude that the urea values obtained did not diverge between the groups under study. The values obtained are in line with those expected for Wistar rats with normal metabolism, more specifically normal renal function [49]. As such, the administration of the extract and the nanoformulation did not compromise renal function.

To evaluate the potential toxicity of the tested formulations, the tissue index values of the collected organs were determined, and the results are depicted in Table 8. It can be observed that the tissue index values for the main organs of the control group and for the tested groups were similar, suggesting the absence of any toxicological effect on the main organs. Furthermore, the safety of the formulations was confirmed by histological analysis (Figure 10), showing no significant changes in the liver, kidney, and spleen of the animals between the tested groups and the control.

### 2.7. Preliminary In Vivo Metabolic Effects Assessed via LC-QTOF-MS

To understand whether the administered free extract (EEB), the nanoformulation (EEB NPs), and ezetimibe (Ezet) affected the cellular metabolism of the animals used in the in vivo assay, particularly cholesterol metabolism and associated metabolites, an untargeted metabolomic analysis of the serum via LC-QTOF-MS was performed.

As commonly used methods for the analysis of metabolic data [51], unsupervised and supervised multivariate analyses were performed, with both exploring changes in metabolites between groups. Using dimensionality reduction methods, such as projection on latent structures (PLS) (Appendix A) and principal component analysis (PCA) (Appendix A), it was possible to identify similarities and differences between the serum of the treated animals relative to the control rats based on the number and intensity of metabolites identified by their *m*/*z* values in the samples [52].

A statistical analysis was also applied to assess whether the detected differences in metabolites were statistically significant (*p* < 0.05) between the compared groups by using Volcano plots of log fold changes for the metabolites from the control and the treated animal serum (Appendix A). The abundant metabolites in serum from both groups, that were identified using the databases PubChem and HMDB, are shown in Figure 11.

Figure 11 represents the analysis of the metabolites detected in the serum of the control rats relative to the metabolites detected in the serum of the rats treated for 4 weeks with EEB and EEB-loaded BSA NPs (EEB NPs), as well as those treated for 5 weeks with ezetimibe (Ezet).

As for the differentially abundant metabolites seen in extract (EEB)-treated serum relative to the control (Figure 11a), an increase in spermidine and sphingosine phosphate was detected. Spermidine is a polyamine, that has already been identified in the *Eisenia bicyclis* seaweed [53], which has antioxidant and anti-inflammatory properties [54] and for which the ability to reduce total cholesterol and LDL-C while increasing HDL-C has also been reported [55]. As for amino alcohol sphingosine phosphate, a component of sphingolipids, it is often associated with an increase in HDL-C [56,57]. Both these metabolite modifications may justify the reduction in total cholesterol observed in extract-treated rats and be associated with the reduction in cholesterol permeation seen in the in vitro assay using the Caco-2 cell model of the intestinal lining. It is also noticeable that, in the case of rats treated with free extract (EEB), the serum contains less L-tryptophan, L-isoleucine, and L-phenylalanine, which are essential amino acids [58], and less of other non-essential amino acids such as L-tyrosine and L-proline, which may lead to modifications in the metabolic pathways of proteins, hormones, or peptide neurotransmitters [59] in these rats.

In the case of the different metabolites detected in the serum of rats treated for 4 weeks with the nanoformulation (EEB NPs) (Figure 11b), the main metabolites observed as being modified are lipids, phospholipids, and fatty acids. Rats treated with the EEB-loaded BSA NPs were found to have an increase in several serum lipids relative to the control rats, notably a higher amount of myristic acid (a fatty acid associated with the decrease in HDL-C [60]), palmitic acid (a fatty acid associated with an increase in LDL-C [61]), N-acylethanolamine (a phospholipid-derived signaling molecule that is associated with a reduction in CVD risk [62]), palmitoyl-glycerophosphocholine (a lysophospholipid associated with total cholesterol and LDL-C decreases and HDL-C increases [63]), and retinal (a diterpene derived from vitamin A that, when insufficient, is associated with cardiovascular risk factors [64,65]), of which seaweeds are considered a good source [43]. Again, in this case, the increased presence of N-acylethanolamine, palmitoyl-glycerophosphocholine, and retinal corroborate the reduction in total cholesterol seen in these rats treated with NPs after 4 weeks in the in vivo assay, which may be on account of the reduction in cholesterol permeation detected in the in vitro assay. In contrast, in these nanoformulation-treated rats’ serum occurred a decrease in stearoylcarnitine, an acylcarnitine associated with inhibition of lecithin:cholesterol acyltransferase (LCAT), an enzyme responsible for the homeostasis of free cholesterol between peripheral tissues and HDL particles [66], and L-isoleucine, an amino acid also found to decrease in the serum of rats treated only with *Eisenia bicyclis* extract.

Finally, it was seen that, in the case of the different metabolites detected in the serum of rats treated for 5 weeks with the drug ezetimibe (Ezet) (Figure 11c), the main metabolites which were modified were lipids, phospholipids, and fatty acids, as was the case with rats treated with the nanoformulation. Rats treated with ezetimibe were found to have, relative to the control rats, an increase in myristic acid (a fatty acid also found to be increased in nanoformulation-treated rats), sphingosine (increased as in the serum of extract (EEB)-treated rats and associated with a decrease in total cholesterol [67]), and 1-O-(2-methoxy-hexadecyl)-glycerol (an alkylglycerol often associated with a decrease in total cholesterol [68]). The increased presence of sphingosine and 1-O-(2-methoxyhexadecyl)-glycerol may account for ezetimibe’s effectiveness in reducing cholesterol permeation, as quantified in the Caco-2 model of the intestinal barrier. In contrast, the drug treatment (Ezet) decreased the levels of two metabolites relative to the control rat serum and oppositely to the other groups (the free (EEB) and encapsulated (EEB NPs) extract-treated groups), namely N-acylethanolamine and retinal, a fatty acid and a vitamin A derivative which may be related to lipid metabolism and a decreased absorption of vitamin A, respectively, that, in some conditions, may occur in association with drug treatments, although nothing was reported associated with ezetimibe. Vitamin A and carotenoid absorption in the intestinal enterocytes is a very complex process involving several transport proteins, amongst which is the Niemann-Pick C1-like 1 (NPC1L1) cholesterol transporter [69], the target of ezetimibe. This decrease in retinal could be related to this effect. However, as several disorders could be associated to vitamin A deficiency and in rats these were associated to problems similar to neurodegenerative disorders [70], this consequence of ezetimibe treatment should be further explored.

## 3. Discussion

Seas are a great source of interesting marine organisms, such as seaweeds with bioactive compounds. However, some of these compounds may require a suitable carrier to be administered. The literature suggests that encapsulation into NPs increases the efficacy of extract compounds [71], or in some cases improves oral absorption through the intestinal barrier [72]. Regarding this route of administration, it is known that oral administration has been the most common therapeutic regimen in various diseases because of its high safety, convenience, low costs, and high patient compliance. However, susceptible to hostile gastrointestinal environments, many drugs show poor permeability across GI tract mucus and intestinal epithelium, with ultimately poor oral absorption and limited therapeutic efficacy.

This work proposes a new oral marine-sourced extract from the seaweed *Eisenia bicyclis* which is loaded in BSA NPs. In terms of the characterization of those NPs, the size of BSA NPs improved after the successful encapsulation of EEB. The high values of encapsulation efficiency (EE) also confirmed our hypothesis. All BSA NPs prepared exhibited a mean size below 300 nm. In fact, size is a key factor for oral absorption. In the absence of aggregation or in the case of small particles, NPs can be more easily absorbed by intestinal epithelial cells. It has been reported that a particle size smaller than 300 nm can prevent enterocytes from taking in substances, while a size larger than 500 nm is more likely to be absorbed through Peyer’s patches [73]. Besides this, the particle size determines the curvature of the NPs [74]. Thus, the shape has also a great influence on their interaction with biological systems, including cellular uptake, plasma circulation, and organ distribution.

Additionally, surface charge plays an important role not only in terms of the physical stability of NPs, but also in terms of the cell interaction and clearance of NPs. It is known that BSA is a negatively charged molecule. Albumin is an acidic, very soluble protein (up to 40% *w*/*v*) at pH 7.4. It is stable in the pH range of 4 to 9. It is described that positively charged particles (zeta potential higher than +10 mV) are known to form aggregates in the presence of the negatively charged serum proteins after intravenous administration [75]. These aggregates are large and often suffer from rapid blood clearance. Researchers also found that positive NPs generally generate a higher immune response (complement activation and conjugate activation) compared to neutral or negative NPs (−40 mV) [76]. On the other hand, negative NPs exhibit stronger reticuloendothelial system (RES) uptake compared to neutral NPs (close to zero, +/−10 mV). These neutral NPs have been associated with much lower opsonization rates than charged NPs with, consequently, lower clearance from the blood circulation [76].

In terms of biological activities, antioxidant activity (AA) reflects the ability to eliminate free radicals, which are harmful for biologic systems because they have the capacity to damage cellular constituents such as proteins, DNA, and lipids [77]. EEB presented very high AA, which was not a surprise since seaweed extracts are known to show beneficial biological effects, often associated to their bioactive compounds’ capacity to neutralize free radicals [39]. However, the AA of EEB was lower after encapsulation. This reduction was already observed after encapsulation of plant extracts in poly(lactic-*co*-glycolic acid) (PLGA) NPs [21], probably meaning that the compounds responsible for the biological effects were not completely exposed due to the reduced reaction time of this assay (very fast assay ≈ 30 min).

Hypercholesterolemia is a major risk factor for the development of CVDs [37]. Regarding hypercholesterolemia, in the present study, we also have observed that both EEB and the EEB-loaded BSA NPs inhibit the HMGR enzyme and reduce cholesterol intestinal permeation, and the results were comparable to commercial and clinically used drugs such as statins (pravastatin) and ezetimibe (Ezet). These drugs act according to different mechanisms: the former by targeting HMGR, the main enzyme in endogenous cholesterol biosynthesis in the liver, and the later by inhibiting dietary cholesterol intestinal absorption [78,79,80]. We here have proved that EEB and the nanoformulation of BSA containing the extract were capable of inhibiting the HMGR enzyme, both showing a similar effect to pravastatin. As the literature suggests and as the results have demonstrated, both the extract and the prepared NPs permeate biological barriers, particularly the intestinal lining. Both the extract and the nanoformulation may be efficient at targeting HMGR in the liver.

Data observed for EEB, in a free form or encapsulated in BSA NPs, were also similar to those registered for ezetimibe [81]. Ezetimibe acts by inhibiting the NPC1L1 protein, selectively preventing the absorption of dietary cholesterol by enterocytes [80,81]. EEB also was seen to decrease the permeation of the cholesterol through a simulated intestinal barrier, especially after encapsulation in BSA NPs. Cholesterol permeation may be affected due to the competition between bioactive compounds for membrane transporters, such as NPC1L1, which transport cholesterol into the intestinal cell, or increase efflux from inside the cells through ATP Binding Cassette Subfamily G Members 5 and 8, known as ABCG5 and ABCG8, respectively. The results reported here are also in agreement with previous in vitro studies that demonstrated that other natural marine products have the capacity to lower the level of plasma and hepatic cholesterol [39].

This work additionally supports the potential of EEB by proceeding from in vitro results to in vivo experiments. In both sets of experiments, there was no mortality among the Wistar rats used in this trial. Weight variation was negligible, which is also a good indicator of animal welfare. Histological analysis and tissue indexes supported the last observation. Specifically, organ weight is one of the most sensitive drug toxicity indicators, and its changes often precede morphological changes [82]. During all assays, animals were normoglycemic since they were healthy and always with free access to feed. In both in vivo assays, it was observed that all the conditions showed the ability to reduce total cholesterol in serum when compared to the control group.

With regard to metabolomics analysis, the administration of EEB, EEB-loaded BSA NPs, and ezetimibe revealed similar metabolic changes in the serum relative to the control group, specifically in terms of amino acids, lipids, phospholipids, and fatty acids, which were associated with lipid homeostasis. These changes additionally confirmed the in vitro results, allowing us to conclude that the extract, and moreover the BSA NPs containing the extract, is effective in treating cholesterol and lipid disorders as its mechanism of action is comparable to statins and ezetimibe. Moreover, in both sets, it was verified that both the EEB and the EEB nanoformulation were safe. Both sets showed that EEB administration did not compromise kidney function or normal organ and body function, thus confirming the safety of their use in monotherapy or in combination therapy with other synthetic and widely used clinical drugs.

## 4. Materials and Methods

### 4.1. Chemicals

Dulbecco’s Modified Eagle Medium (DMEM), Roswell Park Memorial Institute (RPMI) medium, glutamine (CAS No.—56-85-9), and trypsin (CAS No.—9002-07-7) were purchased from Lonza (Verviers, Belgium). Antimycotic phloroglucinol (CAS No.—108-73-6), Folin–Ciocalteu’s phenol reagent, sodium acetate (CAS No.—127-09-3), 2,2-Diphenyl-1-picrylhydrazyl (DPPH) (CAS No.—1898-66-4), cholesterol (CAS No.—57-88-5), acetylcholinesterase (AChE) (CAS No.—9000-81-1), acetylcholine iodide (AChI) (CAS No.—2260-50-6), and bovine serum albumin (BSA) (CAS No.—9048-46-8) were obtained from Sigma-Aldrich (Barcelona, Spain). Sulfuric acid (CAS No.—7664-93-9) and Tris(hydroxymethyl)aminomethane (Tris) (CAS No.—77-86-1) were obtained from Merck kGaA (Darmstadt, Germany). Trifluoroacetic acid (TFA) (CAS No.—76-05-1) and sodium chloride (CAS No.—7647-14-5) were purchased from PanReac (Barcelona, Spain). Methanol (CAS No.—67-56-1), citric acid (CAS No.—77-92-9), magnesium chloride hexahydrate (MgCl2⋅6H2O) (CAS No.—7791-18-6), and ethanol (CAS No.—64-17-5) were obtained from Honeywell (Charlotte, NC, USA). Methanol 96% (CAS No.—67-56-1) was purchased from LabChem (Zelienople, PA, USA). Fetal bovine serum (FBS) (CAS No.—1943609-65-1) was purchased from Biowest (Nuaillé, France). Phosphate buffer saline (PBS) was purchased from Corning (New York, NY, USA). Polygalacturonic acid (CAS No.—25990-10-7), 5-5′-Dithiobis (2-nitrobenzoic acid) (DTNB) (CAS No.—69-78-3), acetonitrile LC/MS grade Optima (CAS No.—75-05-8), formic acid LC/MS grade Optima 64-18-6, isopropanol LC/MS grade Optima (CAS No.—67-63-0), and glacial acetic acid (HPLC) 64-19-7 and AR-certified sodium hydroxide (CAS No.—1310-73-2) were obtained from Thermo Fisher Scientific (Waltham, MA, USA). Phenol (CAS No.—108-95-2) and 3-(4,5-Dimethylthiazol-2-yl)-2,5-diphenyltetrazolium bromide (MTT) (CAS No.—57360-69-7) were obtained from VWR (Radnor, PA, USA). Acetonitrile (ACN) (CAS No.—75-05-8) was obtained from CARLO ERBA (Cornaredo, Italy). Sodium hydroxide (CAS No.—1310-73-2) was purchased from Scharlau (Barcelona, Spain). Ezetimibe (Ezet) (CAS No.—163222-33-1) was obtained from the Azevedos group (Amadora, Portugal). Isoflurane (CAS No.—26675-46-7) was obtained from IsoVet, B Braun (Melsungen, Germany).

### 4.2. Algae Material

The dry *Eisenia bicyclis* seaweed was purchased from a commercial surface from the Seara brand (Batch number T20220405, expiration date April 2022).

### 4.3. Aqueous Eisenia bicyclis Extract (EEB) Preparation

The EEB was prepared with distilled water (33 g/L). The solution was heated in the Uniclave 99 (A.J. Costa (Irmãos), Agualva-Cacém, Portugal) autoclave at 121 °C for 15 min and then put in a cold chamber at 4 °C until it cooled. It was then filtered using Whatman filter paper number 1 and gauze, in order to recover as much extract as possible. After that, the solution was placed in the freezer at −20 °C and dried in the Heto^®^ PowerDry LL3000 (Thermo Fisher Scientific, Milford, OH, USA) freeze dryer.

### 4.4. Chemical Analysis with the HPLC-DAD

High-performance liquid chromatographic (HPLC) analysis was carried out in an Elite LaChrom^®^ VWR Hitachi liquid chromatograph (Tokyo, Japan) equipped with the Column oven L-2300 and Diode array detector L-2455 (VWR, Radnor, PA, USA). A column LiChroCART^®^ 250-4 LiChrospher^®^ 100 RP-18 (5 µm) was used. The samples in the study were analyzed by injecting 25 µL with an auto injector and using a gradient composed of 0.05% trifluoroacetic acid (solution A) and acetonitrile (solution B) as follows: 0 min, 100% A; 30 min, 70% A, 30% B; 40 min, 20% A, 80% B; 45 min, 20% A, 80% B; 50 min 70% A, 30% B; 52 min, 100% A; 55 min 100% A. The flow rate was 0.8 mL/min and the detections were carried out between 200 and 600 nm using a diode-array detector (DAD). The chromatograms concerning determination of the encapsulation efficiency (EE) of the NPs were performed between 13 and 17 min, considering maximum intensity of 250–600 nm. In the gastrointestinal barrier permeation assay, the quantification of the bioactive compounds present in the samples was performed between 7 and 8 min, considering maximum absorbance intensity of 250–600 nm.

The quantification of cholesterol for the permeation study was performed with the HPLC-DAD using a LiChroCART^®^ RP-8, 100 Å, 250 × 4 mm, 5 μm column (Merck) with a mobile phase consisting of a binary system of acetonitrile and methanol while in isocratic mode (50:50, 0–15 min) with a flow rate of 1 mL/min. In each run, 75 μL of sample was injected, with detection between 200 and 400 nm, and the method lasted 15 min. The cholesterol permeation assay was performed between 5 and 6 min, considering maximum absorbance intensity at a wavelength of 210 nm.

### 4.5. Quantification of Total Phenols

Total phenol content was determined using a method described by Oktay et al. [83]. Absorbance was measured at 750 nm using the Folin–Ciocalteu reagent. The standard curve (y = 6.125x + 0.0207; R2 = 0.9755) was obtained with serial phloroglucinol concentrations (0–5 mg/mL). The quantity of total phenol content was calculated as milligrams of phloroglucinol equivalents (PGE) per milligrams of dry EEB as the mean of three replicates.

### 4.6. Quantification of Total Polysaccharides

For polysaccharide quantification, the phenol–sulfuric acid method as described by Masuko et al. [84] was used. Absorbance was measured at 490 nm using polygalacturonic acid. The standard curve (y = 9515x + 601; R2 = 0.9936) was obtained. The quantity of total polysaccharides was calculated as milligrams of polygalacturonic acid equivalents (PE) per milligram of dry mass.

### 4.7. Quantification of Total Proteins

For the quantification of total proteins, the 2-D Quant Kit from GE Healthcare^®^ (Chicago, IL, USA) was used and the standard procedure was followed [85]. To determine the quantity of total proteins, a calibration curve was obtained at 480 nm (y = −0.007x + 0.857) using milligrams of BSA protein per milligram of dry mass.

### 4.8. Preparation of EEB-Loaded BSA NPs

The preparation of the BSA NPs was based on a previous study conducted by Santos-Rebelo et al. [31] with some modifications. In the first step, 100 mg of BSA was dissolved in 4 mL of Milli-Q H_2_O and the pH was adjusted from 7 to 10 (Metrohm© 744 pH Meter, Barendrecht, the Netherlands) with a 0.1 mM NaOH solution (final pH of 8.5). Next, this solution was added dropwise to 16 mL of absolute ethanol under magnetic stirring at 500 rpm (2 mag magnetic and motion MIXdrive 15, Munich, Germany). Subsequently, glucose (1.175 μL glucose/mg BSA) was added to the previous solution under magnetic stirring (500 rpm). This reaction was maintained for 30 min. The BSA NPs were stored at −4 °C for further characterization. For the EEB-loaded BSA NPs, free extract (ranging from 10 to 50 mg) was added to the BSA solution.

Subsequently, the physical characterization of the obtained BSA NPs was performed in terms of average particle size and polydispersity index by dynamic light scattering (DLS) (Zetasizer Nano S; Malvern Instruments, Malvern, UK) in aqueous suspension of BSA NPs and in terms of zeta potential using the electrophoretic mobility technique (Zetasizer Nano Z; Malvern Instruments, Malvern, UK). For the last technique, 25 µL of the prepared sample NPs and 1 mL of PBS were used.

To determine encapsulation efficiency (EE) for the BSA NPs, non-encapsulated extract under the presence of supernatant was collected via centrifugation at 3500 rpm for 5 min and analysed using the HPLC method (method in Section 4.4). EE was calculated using Equation (1), where the supernatant E represents the mass of EEB in the supernatant and Initial E is the initial mass of EEB added.
EE (%) = [(Initial E − Supernatant E)/Initial E] × 100(1)

### 4.9. Production Yield (%)

Since NP preparation involves the use of ethanol, which would make the cytotoxicity assays unfeasible, it was necessary to freeze dry the obtained NPs.

NPs were initially frozen at −2 °C for subsequent lyophilization (Heto^®^ PowerDry LL3000, Milford, OH, USA) for 24 h at −60 °C. After the lyophilization process, the NPs were weighed to determine NP production yield (%) using Equation (2).
Production Yield (%) = (Freeze-dried NPs/Initial components used for NPs preparation) × 100(2)

### 4.10. Morphology of BSA NPs

BSA NP morphology was evaluated via scanning electron microscopy (SEM). The suspensions of NPs were fixed in a mixture of 3% paraformaldehyde and 3% glutaraldehyde in PBS at pH 8.5 for 30 min. After centrifugation at 10,000× *g*, the pellets were washed three times in the buffer of the fixative mixture. Subsequently, aliquots (10 μL) of each sample were dispersed over round coverslips, that were previously coated with poly-L-lysine, and attached to the microscope stubs. The samples were then exposed to osmium tetroxide vapors for 15 min, dehydrated in a graded ethanol series, and dried with hexamethyldisilazane. After coating with a thin layer of gold, the samples were observed on a JEOL 5200 LV scanning electron microscope (JEOL Ltd., Tokyo, Japan) at 20 kV and images were acquired digitally.

### 4.11. Biological Properties of EEB and EEB-Loaded BSA NPs

#### 4.11.1. DPPH Radical Scavenging Assay

The antioxidant activity determination was measured via the 2,2-diphenyl-1-picrilhidrazil (DPPH) method described by Falé et al. [86]. The percentage of antioxidant activity (AA) was determined using Equation (3), where Abs _517 nm Control_ is the absorbance at 517 nm of the control and Abs _517 nm Sample_ is the absorbance at 517 nm of the samples:AA (%) = [(Abs _517 nm Control_ − Abs _517 nm Sample_)/Abs _517 nm Control_)] × 100(3)

After, the determination of antioxidant activity, the EC_50_ of the EEB (amount of extract that has an antioxidant activity of 50%), was calculated using Equation (4). The assays were performed in triplicate.
AA (%) = 210.89 × Concentration *_Eisenia bicyclis_* + 4.06(4)

#### 4.11.2. Inhibition of Acetylcholinesterase Activity

To study acetylcholinesterase (AChE) enzymatic activity, a method described by Falé et al., was followed [86] and Equation (5) was applied:AChE inhibition (%) = 100 − (R_s Sample_/ _s Activity 100%_) × 100(5)
where R_s Sample_ is the reaction speed for the sample under study and R_s Activity 100%_ is the reaction speed for the 100% activity of the AChE enzyme.

This assay was performed in triplicate.

#### 4.11.3. Inhibition of 3-Hidroxi-3-metilglutaril-CoA Reductase Activity

For the inhibition of 3-hidroxi-3-metilglutaril-CoA Reductase (HMGR), a Sigma-Aldrich Kit was used [87] and the percentage of HMGR inhibition was calculated using Equation (6).
HMGR inhibition (%) = 100 − (R_s Sample_ /R_s Activity 100%_) × 100(6)
where R_s Sample_ is the reaction speed for the sample under study and R_s Activity 100%_ is the reaction speed for the 100% activity of the HMGR enzyme.

The assay was performed in triplicate.

#### 4.11.4. Reduction of Cholesterol Permeation Assay

The reduction of cholesterol permeation assay was performed using a transwell and the Caco-2 cell line, based on a previous study conducted by Arantes et al. [88]. Briefly, Caco-2 cells were seeded in 12-well transwell plate inserts at a density of 5 × 10^4^ cells/cm^2^. To start the assay, the cells were washed with Hank’s Balanced Salt Solution (HBSS) twice. The solutions used for testing were then placed in the apical chamber. In the basolateral, 1 mL of HBSS was added. In those wells the (1) HBSS control, (2) sample (free EEB or EEB-loaded BSA NPs), (3) sample plus cholesterol, (4) negative control (cholesterol), (5) positive control (ezetimibe), (6) cholesterol plus ezetimibe, and (7) sample plus cholesterol plus Ezet, all in duplicate, were then added. The final concentration of EEB and EEB-loaded BSA NPs was 0.3 mg/mL, cholesterol was 5 mM, and Ezet was 0.1 mM. The cells were in contact with the solutions for 6 h at 37 °C with 5% CO_2_. After this period, the samples were collected and aliquots of 75 µL were analysed using the HPLC-DAD (Section 4.4).

The percentage of cholesterol was calculated by the height of the cholesterol peak, assuming that the cholesterol in the basolateral chamber at 6 h corresponds to 100% permeation.

This assay was performed in triplicate.

#### 4.11.5. Bioactive Compound Analysis

Samples from Section 4.11.4 were analyzed in the HPLC-DAD with 25 µL aliquots (Section 4.4). The percentage of bioactive compounds was calculated by the height of the sample plus the cholesterol peak, assuming that the sample in the basolateral chamber at 6 h corresponds to 100% permeation.

This assay was performed in triplicate.

#### 4.11.6. In Vitro Safety Assay

The human hepatoma cell line (HepG2) (ECACC 85011430) and human colon carcinoma cell line (Caco-2) (ECACC 86010202) were cultured in DMEM and RPMI medium, respectively, and supplemented with 10% and 20% of FBS, respectively, as well as an antimycotic (10,000 units penicillin, 10 mg streptomycin, and 25 µg amphotericin B per mL), in T75 flasks at 37 °C in an atmosphere with 5% CO_2_. The culture medium was changed every 48–72 h.

In the cytotoxicity determination, the 3-(4,5-dimethylthiazol-2-yl)-2,5-diphenyl-tetrazolium bromide (MTT) assay by Mosmann et al. [89] was used. This assay was performed in triplicate. The percentage of cell viability was determined using Equation (7).
Cell Viability (%) = (Abs _Sample_/Abs _Control_) × 100(7)

#### 4.11.7. In Vivo Efficacy and Safety Assay

All animal studies were performed in compliance with the guidelines outlined in the Guide for the Care and Use of Laboratory Animals, in accordance with the nationally (DL 113/2013, 2880/2015, 260/2016 and 1/2019) and internationally (Directive 2010/63/EU) accepted principles for laboratory animal use (the three R’s principles). All animal experiments were reviewed and approved by the DGAV (national authority) and by the Animal Experiment Ethics Committee of the University of Lisboa (ORBEA).

Thirty-two male Wistar rats (Charles River, Barcelona, Spain) were adapted to laboratory conditions for 7 days before the start of the assays and were kept at 22 ± 1 °C with a controlled 12-h light/dark cycle.

In the first set, an in vivo preliminary assay with the free extract was performed with 12 animals, with the extract administered through their drinking water (7 days a week with a 5-week duration). The rats were divided into 4 groups: group 1, negative control dosed with water (*n* = 2); group 2, positive control dosed with Ezet (0.1 mg/mL) (*n* = 3); group 3, test 1 with the free EEB (0.1 mg/mL) (*n* = 4); and group 4, test 2 with a mixture of pure Ezet and EEB (0.1 mg/mL) (*n* = 3).

Animal weight variation, blood glucose (Element NEO strips, OSANG Healthcare, South Korea), and feed consumption were monitored weekly. The mortality rate was also assessed. In addition, urine was also assessed weekly in order to monitor kidney function (Combur strips, URITEST 10 V, URIT Medical Electronic, Guilin, China). As a way of measuring initial cholesterol, a blood sample was collected via cardiac puncture and centrifuged at 3500 rpm for 10 min (BECKMAN GPR Centrifuge, Indianapolis, USA). Serum was obtained and frozen for later analysis. On the last day, the rats were euthanized using isoflurane and a blood sample was collected and centrifuged. The serum was obtained, and final cholesterol and urea were determined. Finally, the tissue index was also determined by weighing the liver, kidney, and spleen using Equation (8).
Tissue Index = [√ (Organ Weight/Body Weight)] × 100(8)

Once the effect of the free extract was evaluated, a second assay set using oral gavage was performed in order to evaluate the effect of EEB-loaded BSA NPs versus free extract in 20 male Wistar rats after daily administration (5 days a week for 4 weeks). The rats were divided into 4 groups: group 1, negative control orally dosed with only water (*n* = 5); group 2, test 1 with the EEB administered via gavage (1 mg/mL) (*n* = 5); group 3, test 2 with a mixture of Ezet and EEB via oral gavage (1 mg/mL) (*n* = 5); and group 4, test 3 with EEB-loaded BSA NPs (1 mg/mL) using the same route of administration (*n* = 5).

On the last day of the trial, in addition to the previous determinations, the excised organs were subjected to histological evaluation. The excised parts were fixed in 10% formalin, embedded in paraffin, and cut into 5 µm sections for subsequent hematoxylin–eosin staining (H&E staining). Sections were examined under a microscope (Olympus BX51, Olympus Corporation, Tokyo, Japan) and images were obtained using a color camera (Olympus U-TV1X-2) and then analyzed with Olympus analySIS^®^ software (Olympus Corporation, Tokyo, Japan).

Blood was used in the final cholesterol and urea analysis, as well as the other part of the metabolomics analysis. For this, the blood was centrifuged at 3500 rpm for 10 min (BECKMAN GPR Centrifuge, Indianapolis, IN, USA) and the serum was stored at −20 °C.

#### 4.11.8. Serum Analysis via LC-QTOF-MS

For the metabolomic analysis of serum, the serum samples from the in vivo assay were first thawed on ice, added to methanol (1:3), and then homogenized in a Heidolph© REAX 2000 vortex (Schwabach, Germany) for 30 s. They were then centrifuged at 13,000× *g* for 15 min at 4 °C (Eppendorf^®^ 5415 D, Hampton, VA, USA). The supernatant was collected, evaporated to dryness using compressed air at room temperature, and stored at −80 °C. Samples were then resuspended with a solution of methanol and ultrapure water (1:1) and centrifuged at 8000× *g* for 5 min at 4 °C.

Chromatographic analyses were performed on a UHPLC Elute autosampler (Bruker, Bremen, Germany) using an Intensity Solo 2 C18 RP column (100 mm × 2.1 mm, 1.8 μm, Bruker, Bremen, Germany) and a volume of 5 μL of sample (auto injector) was injected into the system. Flow rate was set at 0.250 mL/ min and the column was maintained at 35 °C. Each sample was injected a minimum of 3 times (Table 9).

For mass spectrometry, a QTOF Impact II (Bruker, Bremen, Germany) was also used and data were acquired using Data Analysis 4.4 software. The method consisted of MS/MS scans in positive ionization mode. Signals were recorded in the 50–1500 *m*/*z* range. The capillary voltage was set to 4000 V. Dry gas was maintained at 6 L/min at 200 °C. Collision cell energy was set to 10 eV and 20 μL. The internal calibration solution consisted of 250 mL of water, 250 mL of isopropanol, 750 μL of acetic acid, 250 μL of formic acid, and 0.5 mL of 1N of a NaOH solution.

### 4.12. Statistical Analysis

The results obtained in all assays were expressed as mean ± standard deviation (SD), except the results obtained in the in vivo assays that are expressed by mean ± standard error of the mean (S.E.M.). The statistical analysis of all results was also performed using an ANOVA analysis tool in Microsoft^®^ Excel (Microsoft Office 365, Washington, DC, USA at a confidence level of *p* = 0.05). Regarding the statistical analysis and identification of compounds via LC-QTOF-MS, Metaboscape 4.0 (Bruker, MA, USA) was used.

## 5. Conclusions

Ideally, a single and pure extract should be used in the development of a formulation because it is the simplest way, is less time-consuming, and is cheap. Nevertheless, this is not always the most viable or successful approach. When using complex mixtures, attention should be paid to the synergy of different compounds present in the extract, chemical instability, potential low stability in the biological media, and difficulty in reaching the target after administration. As for pharmaceutical dosage forms, some technological processes must be performed in order to obtain a final product that can assure quality, safety, and efficacy. Poor solubility in water and stability have limited the potential application of natural products in medicines and food supplements. Recently, studies have attempted to address those problems using nanocarriers as a very plausible approach. Based on the results presented here, and based on the biological responses, there are clearly two major advantages of using nanocarriers to associate complex extracts: (i) more promising biological properties in the inhibition of the AChE enzyme and (ii) a reduction of cholesterol permeation in vitro when compared to the free extract. These results are supported by previous studies that report that encapsulation of compounds in NPs increases their efficacy over the same molecules in the free form. Besides efficacy, this nanoformulation demonstrated to be safe in vitro and in vivo.

Overall, the obtained results could certainly attract interest in EEB-loaded BSA NPs as a potential nanoproduct in the very near future with regard to the development of therapies for reducing CVD. Nevertheless, this path might be long and depends on advances in knowledge relating to the scale-up process of this nanoformulation, as well as on the regulatory framework of the type of delivery systems that contain these complex matrices.

## Figures and Tables

**Figure 1 marinedrugs-20-00608-f001:**
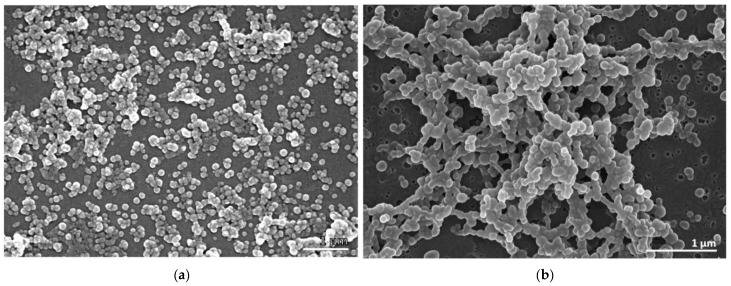
Scanning electron micrographs showing the morphology of the BSA NPs. (**a**) Empty BSA NPs. (**b**) EEB-loaded BSA NPs. JEOL 5200 LV scanning electron microscope (JEOL Ltd., Tokyo, Japan) (Scale bars in white: 1 µm).

**Figure 2 marinedrugs-20-00608-f002:**
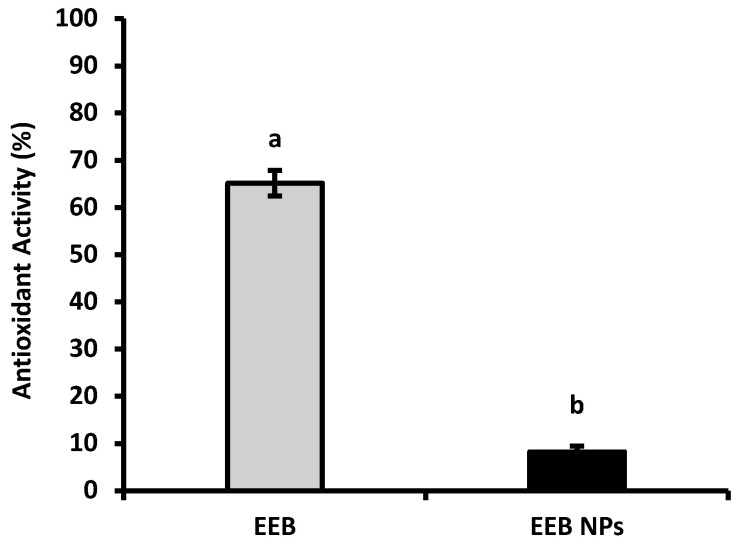
Antioxidant activity of *Eisenia bicyclis* extract in the free form (EEB), and after encapsulation in BSA NPs (EEB NPs), at a concentration of 0.25 mg/mL, with the respective standard deviation. Different superscript letters (a,b) correspond to values that are statistically different between the samples under study (*p* < 0.05).

**Figure 3 marinedrugs-20-00608-f003:**
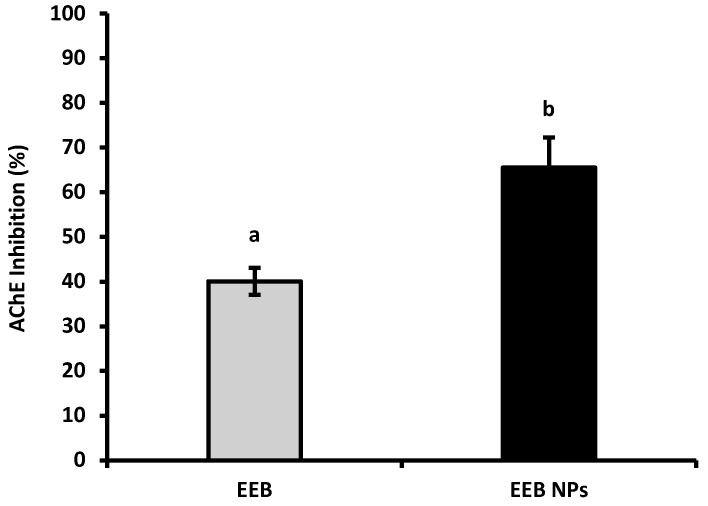
AChE inhibition of *Eisenia bicyclis* extract in the free form (EEB), and after encapsulation in BSA NPs (EEB NPs), at a concentration of 0.25 mg/mL, with the respective standard deviation. Different superscript letters (a,b) correspond to values that are statistically different between the samples under study (*p* < 0.05).

**Figure 4 marinedrugs-20-00608-f004:**
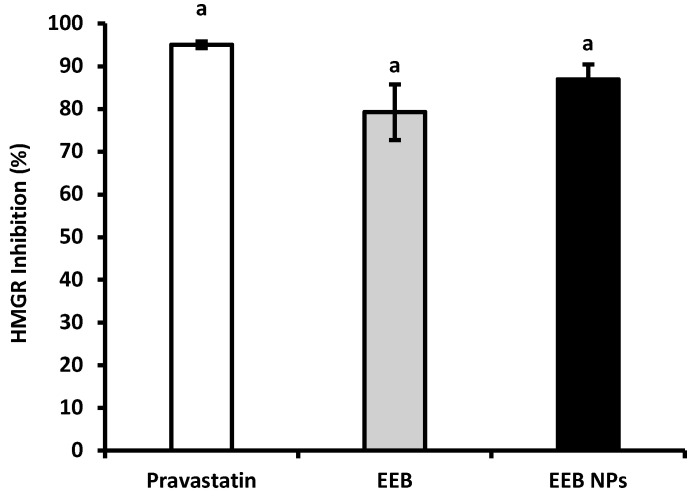
HMGR inhibition of pravastatin, *Eisenia bicyclis* extract in the free form (EEB), and *Eisenia bicyclis* extract encapsulated in BSA NPs (EEB NPs) at a concentration of 0.25 mg/mL with mean values and the respective standard deviation. Equal superscript letters (a) (*p* > 0.05) mean no statistical differences between all groups.

**Figure 5 marinedrugs-20-00608-f005:**
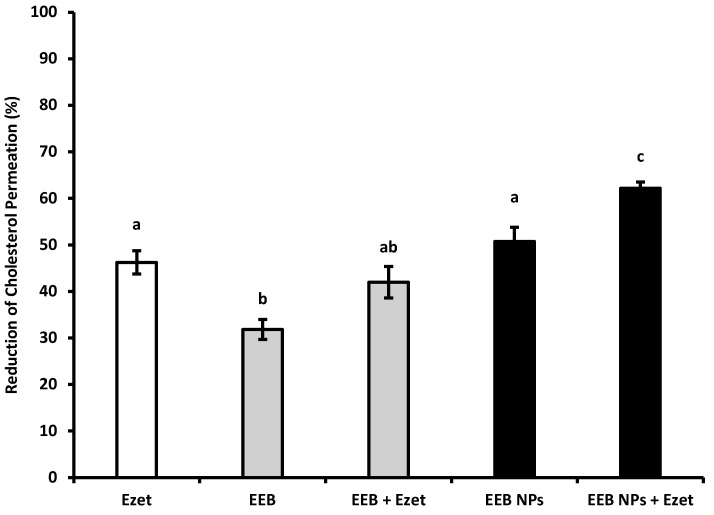
Reduction of cholesterol permeation with 0.1 mM ezetimibe (Ezet), 0.3 mg/mL *Eisenia bicyclis* extract in the free form (EEB), and 0.3 mg/mL *Eisenia bicyclis* extract encapsulated in BSA NPs (EEB NPs) with the respective standard deviation. Human differentiated Caco-2 cells were used in the assay as a model of the intestinal lining and 5 mM of cholesterol (Chol) solution with 6-h permeation to the basolateral chamber was considered as 100% permeation. Different superscript letters (a–c) correspond to values that are statistically different between the samples under study (*p* < 0.05).

**Figure 6 marinedrugs-20-00608-f006:**
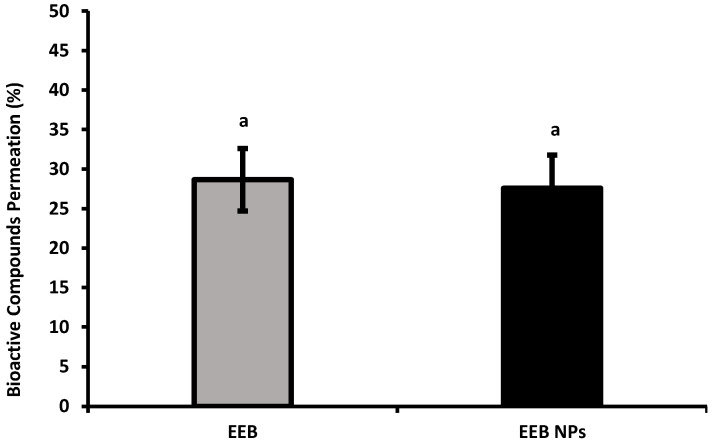
Bioactive compound permeation of *Eisenia bicyclis* extract in the free form (EEB) and encapsulated in BSA NPs (EEB NPs), in the presence of 5 mM cholesterol and 0.1 mM ezetimibe, at a concentration of 0.3 mg/mL with the respective deviation shown. Equal superscript letters (a) (*p* > 0.05) mean no statistical differences between all groups.

**Figure 7 marinedrugs-20-00608-f007:**
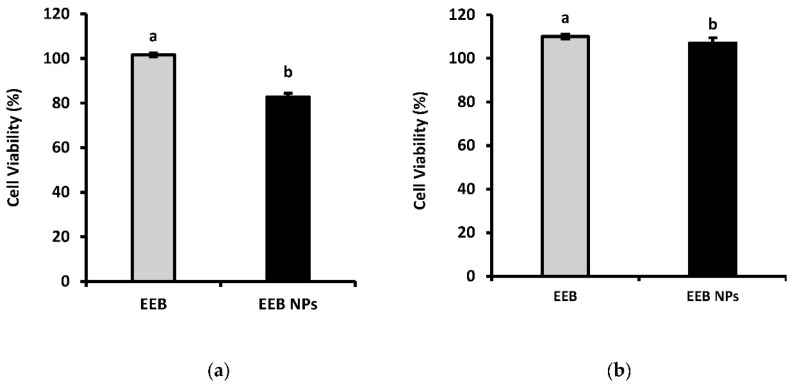
Cell viability (%) of (**a**) Caco-2 and (**b**) HepG2 after 24-h incubation with 1 mg/mL *Eisenia bicyclis* extract in the free form (EEB) and *Eisenia bicyclis* extract encapsulated in BSA NPs (EEB NPs) with mean value and the respective deviation shown. Different superscript letters (a,b) correspond to values that are statistically different between the samples under study (*p* < 0.05).

**Figure 8 marinedrugs-20-00608-f008:**
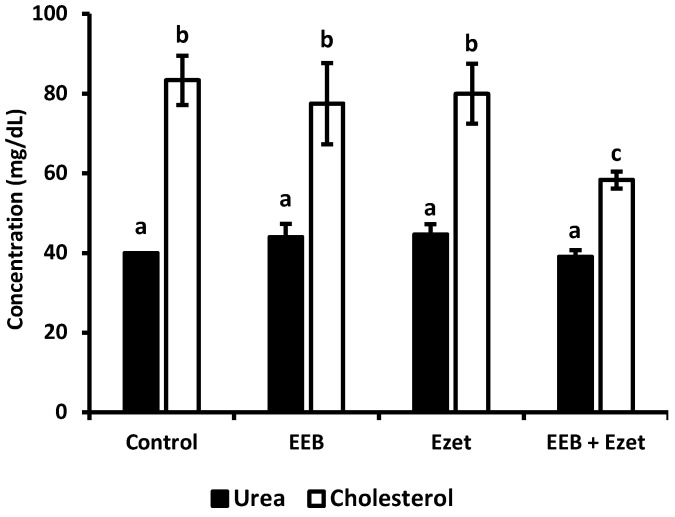
Cholesterol and urea values for each test group after the 5-week period of the in vivo assay where formulations were dosed in the drinking water. Values are presented as relative mean ± S.E.M. Different superscript letters (a–c) correspond to values that are statistically different (*p* < 0.05).

**Figure 9 marinedrugs-20-00608-f009:**
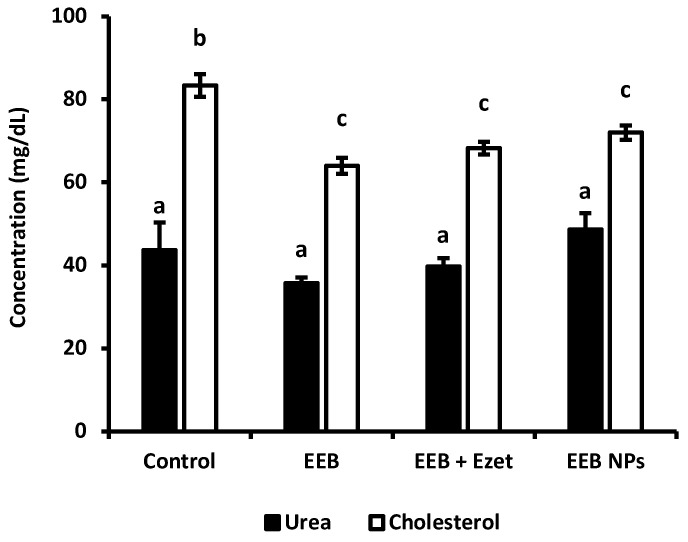
Cholesterol and urea values for each group over the 4 weeks of the in vivo assay with formulations administered by oral gavage (second set of experiments). Values are presented as relative mean ± S.E.M. Different superscript letters (a–c) correspond to values that are statistically different (*p* < 0.05).

**Figure 10 marinedrugs-20-00608-f010:**
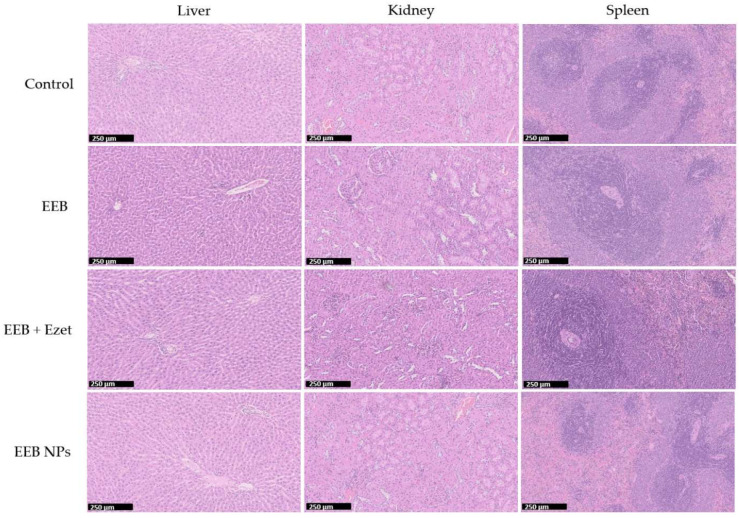
Histological images (H&E staining) of the liver, kidney, and spleen for one rat from each test group over the 4 weeks of the in vivo efficacy and safety assay (second set). Formulations were administered by oral gavage. Scale bar in black: 250 μm.

**Figure 11 marinedrugs-20-00608-f011:**
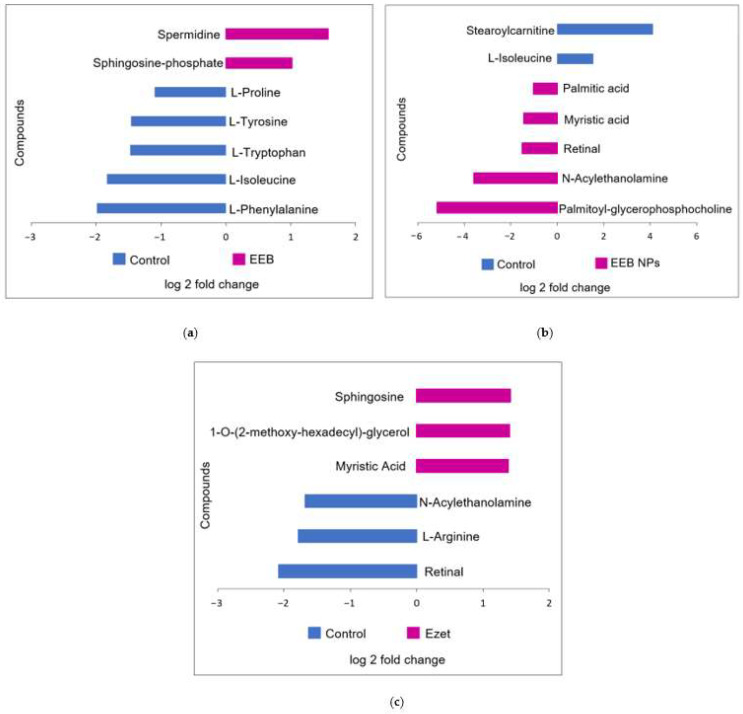
Main identified compounds when (**a**) *Eisenia bicyclis* extract in the free form (EEB), (**b**) *Eisenia bicyclis* extract encapsulated in BSA NPs (EEB NPs), and (**c**) ezetimibe (Ezet) were administered. Results are from one rat in each test group that demonstrated the highest reduction in cholesterol level.

**Table 1 marinedrugs-20-00608-t001:** Size, PdI, and surface charge by zeta potential of EEB-loaded BSA NPs and empty BSA NPs. Values are presented as mean values with the respective standard deviation (SD).

Samples	Mean Size (nm)	Polydispersity Index	Zeta Potential (mV)	EE (%)
Empty BSA NPs	104 ± 8	0.285 ± 0.028	−12.3 ± 2.1	----
EEB-loaded BSA NPs (10 mg of EEB)	226 ± 21	0.471 ± 0.101	−13.0 ± 1.2	96
EEB-loaded BSA NPs (25 mg of EEB)	71 ± 4	0.631 ± 0.181	−11.6 ± 1.2	87
EEB-loaded BSA NPs (50 mg of EEB)	85 ± 7	0.713 ± 0.080	−15.8 ± 4.4	71

NPs: Nanoparticles; EEB: *Eisenia bicyclis* extract.

**Table 2 marinedrugs-20-00608-t002:** Variation of average body weight for each test group over the 5 weeks of the first set of the in vivo efficacy and safety assay. Formulations were administered in the drinking water at a daily dose of 0.013 g/kg b.w. Values are presented as absolute b.w. (g) and, in brackets, as the percentage (%) of relative weight dependent on week 0. Values represent mean ± S.E.M. Differences between groups were evaluated using ANOVA.

Samples	Week 0	Week 1	Week 2	Week 3	Week 4	Week 5
Control	375 ± 18	377 ± 12 (101 ± 2)	394 ± 4 (105 ± 3)	415 ± 6 (111 ± 3)	400 ± 2 (107 ± 4)	432 ± 3 (116 ± 5)
EEB	409 ± 8	418 ± 7 (102 ± 1)	427 ± 6 (104 ± 1)	434± 6 (106 ± 2)	435 ± 10 (107 ± 2)	453 ± 8 (111 ± 3)
Ezet	384 ± 12	391 ± 9 (102 ±1)	402 ± 10 (105 ± 1)	412 ± 10 (108 ± 1)	419 ± 9 (109 ± 1)	440 ± 10 (115 ± 2)
EEB + Ezet	393 ± 6	402 ± 6 (102 ± 0)	412 ± 5 (105 ± 1)	425 ± 5 (108 ± 1)	426 ± 3 (109 ± 1)	447 ± 8 (114 ± 0)

**Table 3 marinedrugs-20-00608-t003:** Variation of average feed intake for each test group over the 5 weeks of the first set of the in vivo efficacy and safety assay. Formulations were administered in the drinking water at a daily dose of 0.013 g/kg b.w. Values are presented as absolute feed intake (g) and, in brackets, as the percentage (%) of relative feed intake dependent on week 1. Values represent mean ± S.E.M. Differences between groups were evaluated using ANOVA.

Samples	Week 1	Week 2	Week 3	Week 4	Week 5
Control	194 ± 1	184 ± 2 (95 ± 1)	187 ± 1 (96 ± 0)	182 ± 3 (94 ± 1)	187 ± 0 (96 ± 1)
EEB	202 ± 4	186 ± 5 (92 ± 2)	177 ± 3 (88 ± 1)	182 ± 5 (90 ± 1)	181 ± 5 (90 ± 2)
Ezet	189 ± 4	176 ± 6 (93 ± 2)	177 ± 3 (94 ± 1)	182 ± 6 (96 ± 2)	181 ± 5 (96 ± 1)
EEB + Ezet	194 ± 1	179 ± 4 (92 ± 2)	177 ± 3 (92 ± 1)	176 ± 3 (91 ± 1)	173 ± 4 (89 ± 1)

**Table 4 marinedrugs-20-00608-t004:** Variation of average glycemia for each test group over the 5 weeks of the first set of the in vivo efficacy and safety assay. Formulations were administered in the drinking water at a daily dose of 0.013 g/kg b.w. Values are presented as absolute glycemia (mg/dL) and, in brackets, as the percentage (%) of relative glycemia dependent on week 0. Values represent mean ± S.E.M. Differences between groups were evaluated using ANOVA.

Samples	Week 0	Week 1	Week 2	Week 3	Week 4	Week 5
Control	139 ± 2	159 ± 1 (115 ± 1)	178 ± 2 (129 ± 1)	164 ± 9 (118 ± 4)	143 ± 11 (103 ± 6)	209 ± 6 (151 ± 2)
EEB	147 ± 1	161 ± 4 (110 ± 2)	172 ± 8 (117 ± 5)	150 ± 5 (103 ± 3)	158 ± 10 (108 ± 7)	174 ± 3 (119 ± 1)
Ezet	149 ± 4	152 ± 4 (102 ± 3)	163 ± 2 (109 ± 4)	143 ± 5 (96 ± 3)	142 ± 2 (96 ± 2)	215 ± 19 (144 ± 14)
EEB + Ezet	147 ± 6	158 ± 5 (107 ± 4)	164 ± 1 (112 ± 4)	156 ± 6 (106 ± 4)	148 ± 3 (101 ± 2)	231 ± 10 (158 ± 12)

**Table 5 marinedrugs-20-00608-t005:** Variation of average body weight for each test group over the 4 weeks of the in vivo efficacy and safety assay with 1 mg of extract in the formulations administered daily by oral gavage. Values are presented as absolute body weight (g) and, in brackets, as the percentage (%) of relative weight dependent on week 0. Values represent mean ± S.E.M. Differences between groups were evaluated using ANOVA.

Samples	Week 0	Week 1	Week 2	Week 3	Week 4
Control	405 ± 7	435 ± 8 (107 ± 1)	427 ± 12 (105 ± 1)	434 ± 12 (107 ± 1)	445 ± 14 (110 ± 2)
EEB	392 ± 18	399 ± 18 (102 ± 1)	400 ± 15 (102 ± 2)	408 ± 15 (104 ± 2)	417 ± 15 (107 ± 1)
EEB + Ezet	428 ± 13	444 ± 14 (104 ± 1)	446 ± 13 (104 ± 0)	456 ± 13 (107 ± 1)	465 ± 13 (109 ± 0)
EEB NPs	411 ± 8	425 ± 7 (106 ± 4)	430 ± 8 (105 ± 0)	437 ± 8 (106 ± 0)	450 ± 8 (110 ± 0)

**Table 6 marinedrugs-20-00608-t006:** Variation of average feed intake for each test group over the 4 weeks of the in vivo efficacy and safety assay with 1 mg of extract in the formulations administered daily by oral gavage. Values are presented as absolute feed intake (g) of five animals from each study group per week. Values in brackets represent the percentage (%) of relative feed intake dependent on week 1. Values represent the mean and differences between groups were evaluated using ANOVA.

Samples	Week 1	Week 2	Week 3	Week 4
Control	801	728 (91)	801 (100)	802 (100)
EEB	862	827 (96)	861 (100)	824 (96)
EEB + Ezet	845	921 (109)	911 (108)	888 (105)
EEB NPs	712	899 (126)	851 (120)	914 (128)

**Table 7 marinedrugs-20-00608-t007:** Variation of glycemia for each test group over the 5 weeks of the in vivo efficacy and safety assay with 1 mg of extract in the formulations administered daily by oral gavage. Values are presented as absolute glycemia (mg/dL) and, in brackets, as the percentage (%) of relative glycemia dependent on week 0. Values represent mean ± S.E.M. Differences between groups were evaluated using ANOVA.

Samples	Week 0	Week 1	Week 2	Week 3	Week 4
Control	76 ± 2	95 ± 9 (125 ± 9)	82 ± 6 (109 ± 7)	94 ± 8 (123 ± 7)	102 ± 7 (138 ± 7)
EEB	76 ± 1	87 ± 3 (115 ± 2)	93 ± 3 (122 ± 3)	88 ± 2 (117 ± 5)	135 ± 16 (179 ± 22)
EEB + Ezet	81 ± 3	87 ± 2 (107 ± 3)	79 ± 2 (97 ± 4)	95 ± 4 (118 ± 6)	128 ± 8 (158 ± 13)
EEB NPs	82 ± 4	84 ± 3 (96 ± 14)	80 ± 8 (97 ± 9)	97 ± 1 (118 ± 6)	115 ± 5 (140 ± 1)

**Table 8 marinedrugs-20-00608-t008:** Tissue index values of the liver, kidney, and spleen for each group over the 4 weeks of the in vivo assay (second set) where formulations were administered by oral gavage. Values are represented as mean ± S.E.M.

	Tissue Index		
Group	Liver	Kidney	Spleen
Control	17.53 ± 0.29	5.70 ± 0.08	4.27 ± 0.08
EEB	16.65 ± 0.23	5.53 ± 0.04	4.01 ± 0.13
EEB + Ezet	16.47 ± 0.37	5.61 ± 0.10	4.01 ± 0.04
EEB NPs	16.92 ± 0.37	5.42 ± 0.11	4.36 ± 0.15

**Table 9 marinedrugs-20-00608-t009:** Method used in LC-MS evaluation.

Time (min)	% Water + 0.1% Formic Acid	% Acetonitrile + 0.1% Formic Acid
0.0	95	5
1.5	95	5
13.5	25	75
18.5	0	100
21.5	0	100
23.5	95	5
30.0	95	5

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
