# Peer review of "Nanoformulation of Seaweed Eisenia bicyclis in Albumin Nanoparticles Targeting Cardiovascular Diseases: In Vitro and In Vivo Evaluation"

_marinedrugs, 2022, doi:10.3390/md20100608_

Round 1

Reviewer 1 Report

Sofia Pinto et al., Nanoformulation of seaweed Eisenia bicyclis in albumin nanoparticles targeting cardiovascular diseases: in vitro and in vivo evaluation. I recommend its publication after making these necessary corrections.

-Minor issues: The abbreviations mentioned in the text, which were not expanded on what they stand for; figures are of poor in quality, try to improve.

-The introduction part of the manuscript should be included "Advantages of Nanoformulation BSA NPs", stand drug side effects, and algae benefit of CVDs (update reference).

-Metal NPs very important functional group; nanocomposites or NPs?, the author should include Eisenia bicyclis FT-IR results. Are the BSA NPs loaded with an extract 6 of Eisenia bicyclis?

- UV results missing the BSA NPs and Eisenia bicyclis separately include

-Missing scale bar and instrument detail in SEM image Fig 1.

-All The statistics information such as (Mean ± SD and the statistical test used to calculate a p-value should be indicated in tables 2,3,4,5,6 and 7.

-Figure 10. Histological images (H&E staining) of liver, kidney, and spleen, scale bar missing and mark changes in pathology

-Could you please provide all chemicals and kits with company name and Cas no.

Author Response

Dear Reviewer

Thank you for the suggestions. Please find attached the answer to the report.

Reviewer 2 Report

The context of the paper should be better described in Introduction. The importance and the emerging area of applications of nanotechnologies on pharmaceutical and nutraceutical science should be marked and related references inserted such as:

Yeung et al. Big impact of nanoparticles: analysis of the most cited nanopharmaceuticals and nanonutraceuticals research, Current Research in Biotechnology, Volume 2, 2020, Pages 53-63, https://doi.org/10.1016/j.crbiot.2020.04.002.

A graphical scheme of study approach should be inserted.

Results on inhibition of acetylcholinesterase enzyme activity should be better described.

Results in Figures 4 and 5 should be better described.

Description on results on In vivo efficacy and safety assay should be implemented.

Limits, advantages and practical applications should be marked in Conclusion.

Author Response

(The authors gave the same response as above.)

Round 2

Reviewer 1 Report

The manuscript  is accepted for publication

Reviewer 2 Report

The paper is suitable for publication